# Analysis of international competitive situation of key core technology in strategic emerging industries: New generation of information technology industry as an example

**Fengyang Wang**, **Zongyuan Huang** *

School of Economics and Management, Beijing Jiaotong University, Beijing, China

* zyhuang@bjtu.edu.cn

## Abstract

In the context of the current technological revolution and unprecedented major changes, countries are facing the situation of accelerating the development of key core technologies, which is caused by the transformation from the dispute over trade to the dispute over ecology and scientific and technological strength. Competitive situation analysis is an important link of key core technology innovation. The construction of a universal model of international competitive situation analysis of key core technology can provide scientific support for decision makers of science and technology innovation to solve technical difficulties. This study takes the new generation of information technology industry as an example, identifies key core technologies of the industry and evaluates the competitive situation of the major world countries. Studies indicate that in the field of new generation information technology, the US and Japan is in the leading position globally. In addition, China has active innovation activities in all fields, but overall there remains a considerable gap with the world-leading level, and its R&D quality needs to be further improved.

**Data Availability Statement:** All relevant data are within the manuscript and its Supporting Information files

**Funding:** The author(s) received no specific funding for this work

## Introduction

Emergence and rapid development of new technologies, such as those in the fields of emerging information technology, intelligent technology, biotechnology, and materials technology, as well as their widespread application in a variety of scenarios [1, 2], have recently changed not only people's lives but also the pattern and structure of society and the economy [3, 4]. The Internet, big data, AI, and 5G are being integrated into the fields of new energy, energy conservation, and environmental protection. These technologies promote the improvement of resource utilization efficiency and the wide application of new technologies for environmental protection [5] and promote green innovation, gradually becoming the key to developing green and low-carbon industries and promoting sustainable environmental development [6, 7]. Therefore, in order to get the upper hand in the race for the future market, the government

**Competing interests:** The authors have declared that no competing interests exist.

and businesses must act quickly to capitalize on the possibilities and trends of emerging technologies, implement efficient R&D plans, and create sensible innovation policies [8].

According to the nature of technology, the key core technology is the key portion of the core technology in the technology system. It is the highest-level technology that accomplishes the core concept and leads assistive technology [9]; it plays an indispensable, vital, leading, and decisive role in the technology chain and industry chain, and it is crucial to breaking the technological restrictions on industrial development and breaking the technological blockade of other countries. According to the history of the industrial revolution, the emergence and catch-up of Britain, the United States, Germany, and Japan, all depended on the backing of emerging key core technologies [10]. Nowadays, With the technological advantages gained during the previous industrial and technological revolutions, these countries now occupy the commanding heights of emerging technologies, dominate technical standards, and form technical barriers, severely limiting the progress of emerging technologies in latecomer countries [11]. As a result, in order to catch up in the present wave of technological revolution, China must further accelerate the speed of key core technologies in strategic emerging industries.

The key core technology comprises characteristics such as large investment and a long period of R&D, complexity and reluctance of knowledge, and industrial ecological reliance, making breakthrough tough and complicated. On the basis of identifying the key core technologies of the industry, evaluating the situation of international competitive of technologies and clarifying the status of research and development in specific fields are important preconditions for determining the direction of a country's key core technologies, which has important practical significance and strategic value for its breakthrough work and layout of key core technologies, improving its scientific and technological innovation ability, and promoting economic and social development.

## Literature review

### Identification of key core technologies

As the mainstream expression of technological information, patents are the carrier and crystallization of technology innovation. More than 90% inventions and creations are covered in patent literature, and nearly 80% technical information can only be obtained in patent literature [12]. Additionally, as patent applications are a type of intellectual property that, to some extent, reflect a nation's technical prowess and level of innovation [13], patent data offers a useful perspective for technology analysis and prediction [14, 15]. The total number of patents in existence today is enormous and increasing quickly, yet there are very few significant patents that are essential to the advancement of the industrial technology system. Therefore, identifying the key core patents from the huge number of patents is critical for comprehending present technology levels and properly anticipating technological development trends [16]. In present research, the following approaches for patent identification are often used: single index method, combination index method, index system method, patent information visualization, and so on.

The citation frequency of patents, patent family size, and quantity of claims are the most frequent single indexes for researchers. The citation frequency of patents is often used to determine and measure the importance of a patent and is significantly and positively correlated with its actual impact. It can be directly used as an important index to identify important patents and key technologies [17]. However, since patent citations usually peak 2 to 4 years after publication, the importance of relatively new patents may be underestimated compared to those older ones [18]. In addition, inaccurate analysis results are produced because it is difficult to identify technical features such as technical importance, technical knowledge flow, and

technical cross-influence from a technical field perspective when only the citation relationship between individual patents is taken into account [19]. A patent family is a group of patents that are applied in different countries with the same or basically the same content, and they have a common priority [20], and all patents in one patent family have the common priority. Because the cost of patent applications increases as the number of countries applying for patent protection increases [21], the more patents of one patent family, the higher its economic and technical value [22]. The quantity of claims in a patent document is crucial because it captures the technical details of the invention as well as the extent of the legal protection that the patent enjoys [23]. According to the patent law, the substance of a patent's claims determines the extent of its protection. In order to ascertain the extent of an invention or utility model patent's protection, the number of claims is essential. The more essential the patent, the more claims it asks for protection against since in the majority of nations the patent applicant must pay additional costs for claims exceeding the allowed number [24, 25].

Combination of indexes means combining several common single indexes to improve the overall identification results by combining the results of several indexes. The combination of indexes can reduce the one-sidedness of a single index to a certain extent, but it ignores the differences in the importance of different indexes. The patent index system typically chooses index based on the technical, financial, and legal characteristics of patents [26], assigns weights to the index using the expert method or the Entropy value method, calculates the scores of each patent following the calculation, and then chooses the patents with the highest scores as the core patents [27]. This approach is being utilized more frequently in the study of identifying core patents as a result of the extensive research viewpoint and scientific theoretical model.

Citation networks and co-occurrence networks are commonly used methods for visualizing patent information. The citation network reflects the accumulation and inheritance relationship among technologies, and the key paths of transfer, inheritance, and flow among patents represent the backbone and pulse of technology development [28]. The visualization technology may highlight the trajectory and development direction of technology in a certain sector and visualize the intricate relationships between patents [29, 30], but it exhibits a temporal lag and disregards the motivation of the citation [31]. Co-occurrence network, or Co-word analysis, which is based on the co-occurrence of keywords or subject terms in the patent text, to explore the convergence between technical themes [32], this method is often used for common technology identification and technology convergence measurement [33, 34]. However, due to the lack of semantic information mining, processing synonyms and polysemous terms is difficult [31], and this technique may fail to extract the latent technical topics in the text, leading to information loss.

## Analysis of technology competition situation

Technological competitiveness refers to the comprehensive capability of different countries or organizations in specific technological fields. Technology competitive situation analysis is a thorough understanding of the growth of a particular technology field by a country or an organization, and it is crucial for them to understand the trend of technological advancement, clarify the competitive position, learn about the R&D status and strategic layout of the competitors, and clarify their own positioning, R&D strategy, and future development direction [35, 36]. And it is the fundamental task for the government to formulate innovation policies and for enterprises to choose R&D strategies [37].

At present, the researches on the analysis of technological competitive situation from the perspective of patent focus on: (1) Analysis of technological competition, Li (2015) [38], Huang (2016) [39], Li (2020) [40], Wen (2022) [41] and others researchers reveal the

competitive landscape within the technology field based on indexes such as major rights holders, number of patent families, and collaborative networks, and analyze the R&D strength of competitors. Song (2022) [42] combined LDA and word2vec to identify the main competitors in China's cloud computing field. (2) Regional analysis of technological competition: Karvonen (2016) [43], Wu (2019) [44], Ahn (2020) [45] analyze the patent distribution of the main competing countries/regions in the technological field by counting the number of patent applications in different countries, regions, or organizations or the priority countries/regions of patent applications. (3) Analysis of technological competition hotspots. A large number of studies determine the field of technical research hotspot based on IPC classification number statistics [46]. Besides, some scholars also make use of patent visualization technology to cluster fields such as patent title and abstract and draw patent topographic maps, so as to dig hot spots and blank areas of technology competition [47, 48].

In conclusion, the existing studies provide useful references for this paper; however, most of the existing studies focus on the quantitative characteristics of patents, and their analysis of the technological competition situation stays in the macro perspective of the overall assessment. There is still a lack of research on how to accurately measure the gap between competitors in specific technical fields. In addition, compared with other technologies, the key core technology of strategic emerging industries is a national weapon with more important technical, economic, and strategic values, and it is an important pillar for the secure development of a national economic system. Therefore, on the basis of quantitative characteristics, combined with the basic characteristics of the key core technologies, this paper constructs the key core technology identification index system, identifies the key core technologies of strategic emerging industries, and analyzes their international competitive situation.

## Research methods

This paper proposes a framework for analyzing the international competitive situation of key core technologies in strategic emerging industries, as illustrated in Fig 1.

### Formulation of search strategy

This part mainly determines the scope of industrial technologies according to the industrial reports, technical reports, relevant literature and industrial technical standards of the target industry. Then, determine the search expressions of patents and select the database to obtain the initial patent sets based on the keywords and the patent IPC classification numbers of the industrial technologies. Finally, acquire a valid patent set by data cleaning and screening.

### Identification of key core technologies

In this section, the key core patent identification index system is built, weights are assigned using the Entropy technique, the comprehensive score of the patents is calculated using the TOPSIS method, and the key core patents are then eliminated using a specific patent classification.

**Index system.** Identifying high-quality patents from a large number of patents is a prerequisite for acquiring key core technologies. Strategic emerging industries are sectors based on significant technological breakthroughs and applications, carrying the important mission of breaking through technological bottlenecks in industry development and overcoming developed countries' technological blockades, and the key core technologies in these industries are the driving force and backbone of the development of industries, and have significant impacts on other technologies in the industry technology system, as well as other industrial technology systems. Key core technologies have the following characteristics: (1) Fundamental

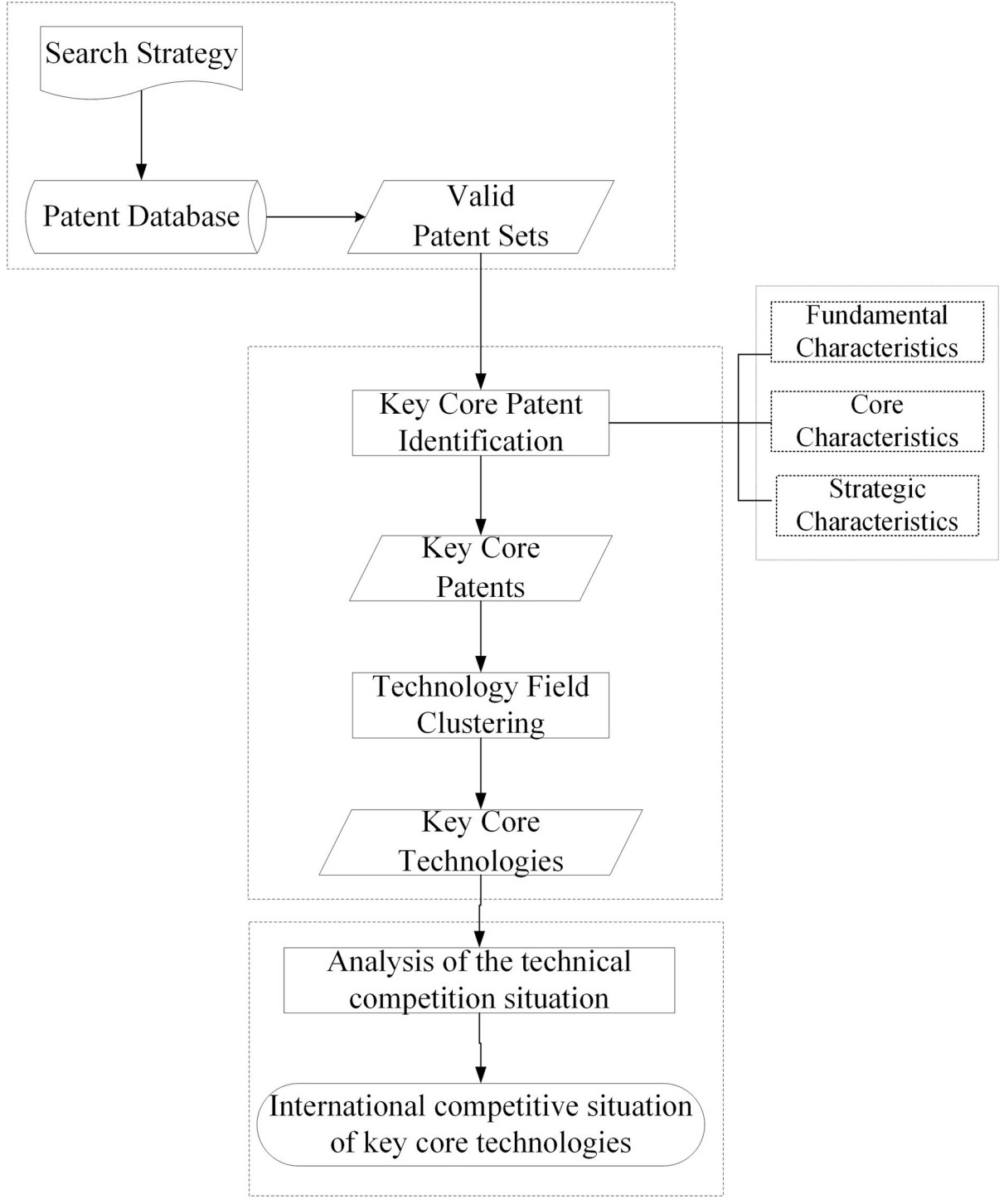

**Fig 1. Analysis flowchart of international competition situation of key core technologies.**

characteristics. Key core technology is the basis and key component of the development of the entire industrial technology system, with high investment and long periodicity, is the core product of scientific research, is a breakthrough original technology, with the difficulty of imitation as well as innovation by the subsequent technologies [49]. (2) Core characteristics. The

key core technology is the highest level of technology in the complex industrial technology system to implement the core concept and lead the auxiliary technology, which plays an indispensable, vital, leading, and decisive role in the industrial technology chain and has the characteristics of irreplaceable, strong influence, and technological dominance. These technologies have broad applications and can influence or even determine the development direction of many technologies in the technology system, and have an overall control role for the track direction of technology development and a significant impact on other technologies [50]. (3) Strategic characteristics. Key core technology is an essential component of the core competitiveness of an industry, which is a scarce, asymmetric, oligopoly technology mastered by a few enterprises or researchers, and the breakthrough of such technology can promote the upgrading, development, and maturity of the industry chain, bringing huge economic and social benefits for the high-quality development and market application of the whole industry, and is of high value [51].

Thus, this paper constructs a key core patent identification index system for strategic emerging industries from three characteristic dimensions: fundamental characteristic, core characteristic and strategic characteristic (see in Table 1).

**Calculation of index weights.** In the past, the index system identification method mainly focuses on subjective weighting, which ignores the law of patent information data and is easy to be affected by personal subjective experience. Therefore, we use the objective weighting method—Entropy method—to determine the weight of each index of the key core technology identification index system, so as to identify the key core patents. The weight of the index determined by the Entropy method, which is now widely used in multi-index comprehensive assessment, is only connected to the variation of the index's real value. The specific calculation steps are as follows:

Firstly, construct the judgment matrix of the key core patent identification index and standardize it. Suppose there are $i$ patent and $j$ identification indexes, $X_{ij}$ ($i = 1,2\ldots, n; j = 1,2,\ldots, m$) representing the value of the patent $i$ on the index $j$, the judgment matrix as shown in

**Table 1. Key core patent identification index system.**

| Index Type | Index | Index Meaning | Index calculation |
|---|---|---|---|
| **Fundamental Characteristic** | Technical Foundations | It shows the technological accumulation of patents. | Number of patents cited |
| | Scientific Foundations | It reflects the patent research basis and research level [52]. | Number of literatures cited |
| | Patent Claims | The higher the number of claims, the greater the scope of technical protection and the higher the value of the patent. | Number of claims |
| **Core Characteristic** | Patent Influence | The more patents cited, the greater the significance of the technical content of the patent for the subsequent innovation, the higher the patent value. | Number of cited patents |
| | Patent Width | The number of patents covering different types of technical subject matter constituting inventive information, indicating the extent of the technical field covered by the patent. | Number of different IPC class numbers in the first four digits |
| | Patent Applications | The greater the number of patent applications, the greater the number of institutions cooperating and the greater the value of the patented technology developed in cooperation [53]. | Number of patent applications |
| **Strategic Characteristic** | Patent Family Size | The number of countries in which the applicant seeks patent protection for the same invention patent | Number of patents in one patent family |
| | National Layout of Patents | The more countries the patent is laid out in, the larger the market scope and the higher the value of the patent strategy [54]. | Number of countries in the patent family |
| | Layout in Developed Countries | Developed countries have a higher level of technology, so the number of patents filed in developed countries can reflect the patent quality to a certain extent. | Number of developed countries in the patent family |

formula (1). Then standardize the judgment matrix as shown in Eq (2).

$$X_{ij} = \begin{bmatrix} X_{11} & \cdots & X_{1m} \\ \vdots & \ddots & \vdots \\ X_{n1} & \cdots & X_{nm} \end{bmatrix} \tag{1}$$

$$X'_{ij} = \frac{X_{ij} - min(X_i)}{max(X_i) - min(X_i)} \tag{2}$$

Secondly, calculate the $P_{ij}$ of patent i under the identification index j by Eq (3), and calculate the entropy value of all identified indexes by Eq (4).

$$P_{ij} = \frac{X_{ij}}{\sum_{i=1}^{n} X_{ij}} \tag{3}$$

$$E_j = -\frac{1}{\ln(n)} * \sum_{i=1}^{n} P_{ij} * \ln\left(P_{ij}\right) \tag{4}$$

Finally, calculate the entropy weight $W_j$ of all identification indexes using Eq (5). The smaller the entropy value is, the more information it can provide and the greater the index weight.

$$W_j = \frac{1 - E_j}{\sum_{j=1}^{m} \left(1 - E_j\right)} \tag{5}$$

**Score of patents.** Each patent is scored by the TOPSIS method, and the key core patents are identified in descending order of score. The fundamental principle of the TOPSIS method is to rank evaluation objects by measuring their distance to the best and worst solutions, and those schemes that are closest to the optimal solution and farthest from the worst solution are regarded as the best of the alternatives. The author believes that the key core patent should be the one whose value tends to be optimal in all aspects, which is consistent with the core concept of the TOPSIS method, so this method is adopted for the identification of key core patents. Here are the steps of the method:

Step 1, Construct a normalized matrix with weights based on the calculation of weights using the Entropy method.

$$V = \left(v_{ij}\right)_{n \times m} = \left(W_j X'_{ij}\right)_{n \times m} \tag{6}$$

Step 2, Determine the optimal solution $V^+$ and the worst solution $V^-$ of the weighted normalized matrix $V_{ij}$.

$$V^+ = \left[v_1^+, v_2^+, \ldots, v_n^+\right] \tag{7}$$

$$V^+ = \left[v_1^+, v_2^+, \ldots, v_n^+\right] \tag{8}$$

And $v^+ = max\left\{v_{1j}, v_{2j}, \ldots, v_{nj}\right\}, v^- = min\left\{v_{1j}, v_{2j}, \ldots, v_{nj}\right\}$

Step 3, Calculate the distance $D_i^+$ between each indicator and the optimal value, as well as the distance $D_i^-$ from the worst value.

$$D_i^+ = \sqrt{\sum \left( v_{ij} - v_j^+ \right)^2} \tag{9}$$

$$D_i^- = \sqrt{\sum \left( v_{ij} - v_j^- \right)^2} \tag{10}$$

Finally, calculate the relative proximity $C_i$ of each unit indicator value to the optimal value.

$$C_i = \frac{D_i^-}{D_i^+ + D_i^-} \tag{11}$$

Ranking evaluation objectives according to proximity. The larger C indicates that it is closer to the optimal level, and the higher the patent score, the stronger its criticality and centrality. Referring to the method of Noh (2016) [55], Park (2016) [56], Yang (2021) [57] and others to select core patents based on the percentage of index score ranking, combined with the comprehensive consideration of the basic characteristics of key core patents, we set the patents with the top 20% score as key core patents.

## Clustering of key core technologies

Based on the International Patent Classification Number (IPC), we use the patent clustering method to construct the Patent-IPC matrix, and identify the key core technologies combined with the network visualization analysis.

The technical field of the patent is expressed in the patent document using the patent classification number. The International Patent Classification adopts a hierarchical technology structure containing five levels: section, class, subclass, main group and subgroup. In the existing studies, technology subclass and main group are widely used to refer to the technical fields of a patent. In order to obtain the key core technology fields more specifically and clearly, we use IPC main group numbers to describe the technical features of key core patents. Patents usually contain multiple patent classification numbers, and patents with the same patent classification number belong to the same technical field. Therefore, based on the IPC classification number of patents, the patents belonging to the same technical field can be clustered through the correlation matrix (see Table 2), and then, we use the network visualization analysis to cluster the obtained key core patents and get the key core technologies. The clustering diagram is shown in Fig 2, in which circular nodes represent patents, square nodes represent technical fields, and lines represent different technical fields to which a patent belongs. The size of nodes represents the degree of the key core of the technology, which is measured by the quantity of key core patents in the corresponding technical field.

**Table 2. Patent-IPC correlation matrix.**

|  | IPC 1 | IPC 2 | IPC 3 | … | IPC n |
|---|---|---|---|---|---|
| **Key Core Patent 1** | 1 | 0 | 1 | … | 1 |
| **Key Core Patent 2** | 0 | 1 | 0 | … | 1 |
| **Key Core Patent 3** | 0 | 1 | 0 | … | 1 |
| … | … | … | … | … | … |
| **Key Core Patent m** | 1 | 1 | 0 | … | 0 |

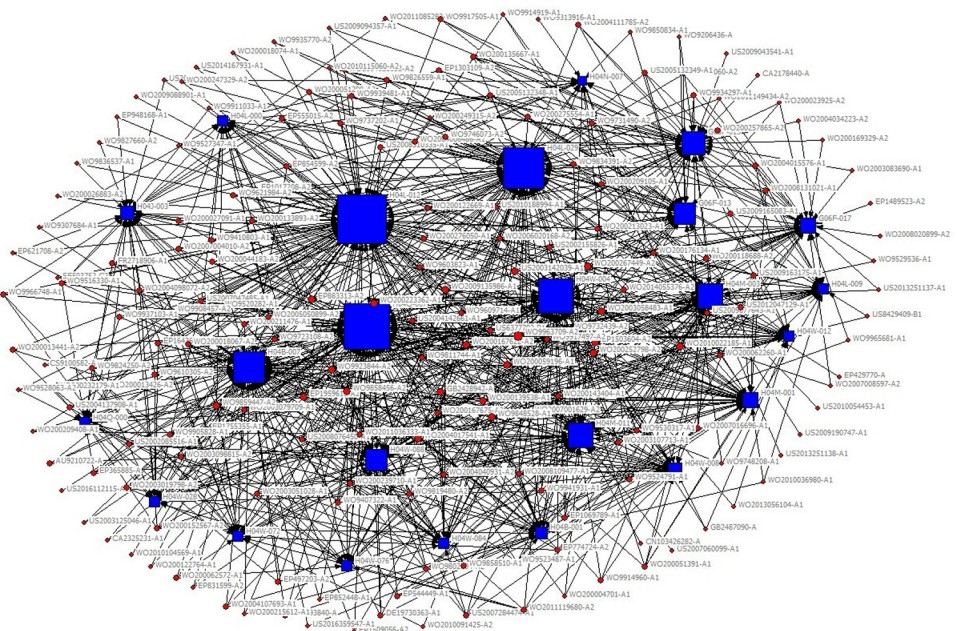

**Fig 2. Clustering diagram of key core technologies.**

## Analysis of technical competition situation

Drawing on existing research, this paper uses a two-dimensional matrix combination analysis to assess the competitive situation in various areas of key core technologies at the national level in terms of both patent quantity and patent quality, where the horizontal axis represents the country's technological innovation activity (represented by the percentage of patent applications in each country) and the vertical axis represents the country's leading technology level (represented by the average score of the top three patents in each country). According to the country's position in the two dimensions, it can be divided into four regions, as shown in Fig 3: technology leaders, potential competitors, technology activists, and technology latecomers.

## Empirical analysis

### Basic overview of new generation of information technology industry

A new technological revolution, which is centered on the new generation of information technologies such as big data, cloud computing, the internet, blockchain, and artificial intelligence, is accelerating and driving sustained and robust growth in the modern economy [58–60]. It has also promoted comprehensive changes in the forms and operating modes of economic and social development [61]. Thus, the new generation of information technology has become a focal point of competition and strategic field as well as the priority direction of development of countries all over the world [62, 63]. In the *Report on Emerging Technology Trends 2016–2045*, the United States regards information technology as the key direction of future science and technology development. *The National Defense Authorization Act for Fiscal Year 2022* approved $14.7 billion in research spending, focusing on the development of microelectronics, artificial intelligence, 5G and other new generation information technologies, and passed a $52 billion *chip bill*. The EU announced in 2020 that it would invest a total of 45 billion euros in chip-related R&D, infrastructure and production by 2030; Japan has set a budget of US $10.7 billion for research and development in 2022, with a focus on AI, big data and quantum

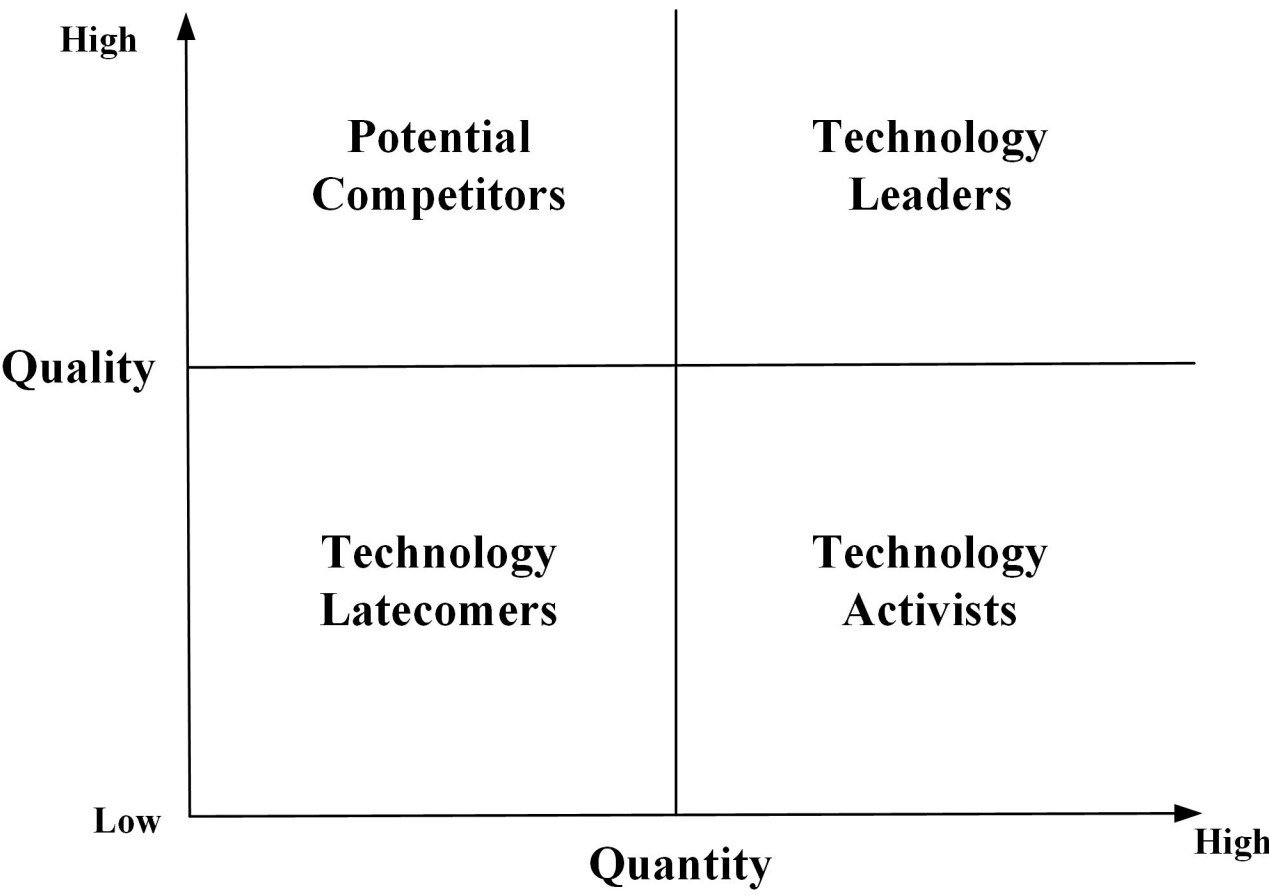

**Fig 3. International competition situation analysis of key core technologies.**

technology. In addition, emerging economies such as Malaysia, Indonesia and Vietnam have also released relevant strategic plans for the layout of next generation information technology to promote the progress of digital economy.

The information technology industry is the general term for information collection, storage, processing, transmission, service and corresponding information equipment manufacturing and service departments. The new generation technology industry intelligently transforms firmware, infrastructure, and service capabilities related to information and networks through research and development as well as the application of new information technologies and equipment. *The Decision of The State Council on Accelerating the Cultivation and Development of Strategic Emerging Industries of China* issued in 2010, pointed out that the new generation of information technology industry focuses on areas such as next generation of communication networks, the Internet of Things, the integration of three networks, new flat displays, high performance integrated circuits and high-level software represented by cloud computing. In 2018, China released the *Classification of Strategic Emerging Industries (2018)*, which explicitly stated that the new generation of information technology industry includes the next generation information network industry, electronic core industry, emerging software and new information technology services, internet and cloud computing big data services, and artificial intelligence industry (see in Table 3). These industries are characterized by high technological content, strong linkage effect, integration, intelligence and application, involving materials, energy, transportation, information, automation and other industrial fields, and their

**Table 3. New generation information technology industry classification and industry scope.**

| Classification | Industry Scope |
|---|---|
| **Next generation information network industry** | Network equipment manufacturing, new computer and information terminal equipment manufacturing, information security equipment manufacturing, new generation of mobile communications network services, other network operation services, computer and auxiliary equipment repair, etc. |
| **Electronic core industry** | New electronic components and equipment manufacturing, electronic special equipment and instrumentation manufacturing, high energy storage and key electronic materials manufacturing, integrated circuit manufacturing, etc. |
| **Emerging software and new information technology services** | Emerging software development, network and information security software development, Internet security services, new information technology services, etc. |
| **Internet and cloud computing, big data services** | Industrial Internet and support services, Internet platform services (Internet +), cloud computing and big data services, Internet-related information services, etc. |
| **Artificial intelligence industry** | Artificial intelligence software development, intelligent consumer-related equipment manufacturing, artificial intelligence system services, etc. |

applications span the three major industries of agriculture, industry and service in the national economy, and their industrial scale and number of leading enterprises are at the top of strategic emerging industries. These industries have low requirements for resources and industrial base, while having a high entry threshold, showing obvious capital-intensive and technology-intensive characteristics.

The front end of the industry chain is electronic devices and communication devices. Electronic devices include electronic components, electronic equipment instruments, electronic materials, integrated circuit manufacturing and other fields. The core is integrated circuit manufacturing [64], involving circuit design, exposure, etching, film growth, package, test and other technologies. The middle end of the industry chain is mainly in the field of communication network, which provides direct support for terminal product applications, including network equipment, communication equipment manufacturing and communication network services, involving computer technology, network technologies such as network architecture, information exchange, access and transmission, as well as communication technologies. The back end of the new generation information technology industry chain emphasizes applications, mainly referring to the application and service industries of modern information technology, such as software services, information services, data services, AI applications and other fields, involving intelligent manufacturing, intelligent transportation and smart phones, computers, TVs and other products.

From the global industry chain division of labor pattern, as shown in Fig 4, the United States is concentrated in the field of integrated circuit and production equipment manufacturing, Japan is concentrated in materials and production equipment manufacturing, mainland China is currently concentrated in integrated circuit packaging and testing, data communication equipment manufacturing and downstream processing and manufacturing of end products. In recent years, the development momentum of China's new generation of information technology industry has been strong, regional industrial clusters have been formed in the in the Yangtze River Delta, Pearl River Delta, Bohai Sea and the central and western regional, with the national IT industry base and the national new generation IT industrial Park as the main parts. According to the *China Strategic Emerging Industries Development Report 2022*, in 2022, the revenue scale of China's new generation information technology industry reached

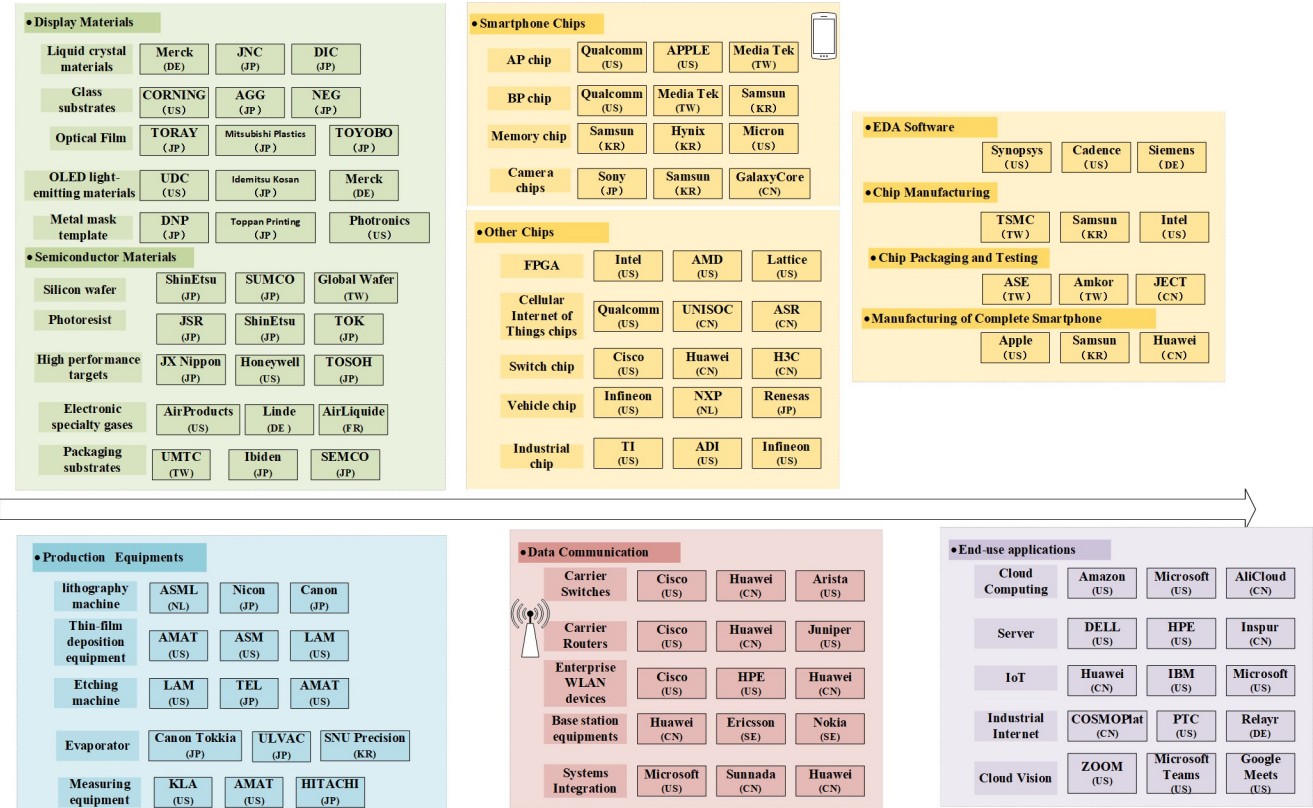

**Fig 4. Global pattern of new generation emerging technology industry chain.**

RMB 28.8 trillion, 7.8% year-on-year growth, accounting for about one third of GDP (RMB121.02 trillion) and three times that of RMB 7.8 trillion in 2010.

## Analysis of the development trend of new generation information technology

To ensure the reliability of data acquisition, we choose the Derwent Innovation Index(DII) database as the data source. The Derwent Innovation Index database collects patent literature and citation information from more than 50 patent offices around the world, with 6,000 records updated every week. Currently, it has collected more than 100 million patent documents around the world. It is bidirectionally connected with Web of Science to connect basic research results and technology application results, ensuring the comprehensiveness and reliability of the data. It is one of the authoritative databases of patent information and science and technology information.

The authors constructed patent search formulas for new-generation information technology industries based on the *Strategic Emerging Industry Classification and International Patent Classification Reference Table (2021)* issued by the China Intellectual Property Office, as well as relevant technical reports and literature, and search the patents on DII. The search period is from 1990-01-01 to 2022-12-31, and the search time is March 2023. The search results show that, there are 554,782 patent families of the next generation information network industry, the electronic core industry 547,838, the emerging software and new information technology

services 95,230, the internet and cloud computing and big data services 237,635, and the artificial intelligence industry 58,327. The trend of patent applications is shown in Fig 5.

By Fig 5, it can be seen that: (1) The patent applications in the next generation information network industry rose gently over the years before 1995, indicating that this stage was the early stage of the growth of the next generation information network technology. From 1996 to 2009, the next generation information network technology developed rapidly in the process of continuous exploration and research. Since 2010, with the continuous development of technology, as it becomes more and more mature, the amount of patent applications has shown a fluctuating growth trend. (2) Prior to 2002, the amount of patent application in the electronic core industry rose gently, then, with the continuous realization of technological breakthroughs, many countries and enterprises actively carry out research and development, and the number of patent applications shows a continuous growth trend. Since 2011, with the continuous release of the potential of integrated circuit technology, many countries have further promoted technology research and development, and the speed of patent application has been further accelerated. (3) Before 2005, there were relatively few patent applications for the emerging software and new information technology services, and the growth was relatively gentle, and the overall activity was relatively low, indicating that relevant technologies were still in the process of continuous exploration and research. After 2006, the number of patent applications began to accelerate, especially since 2015, when the speed of patent applications further accelerated, and related technologies developed rapidly. (4) Before the 21st century,

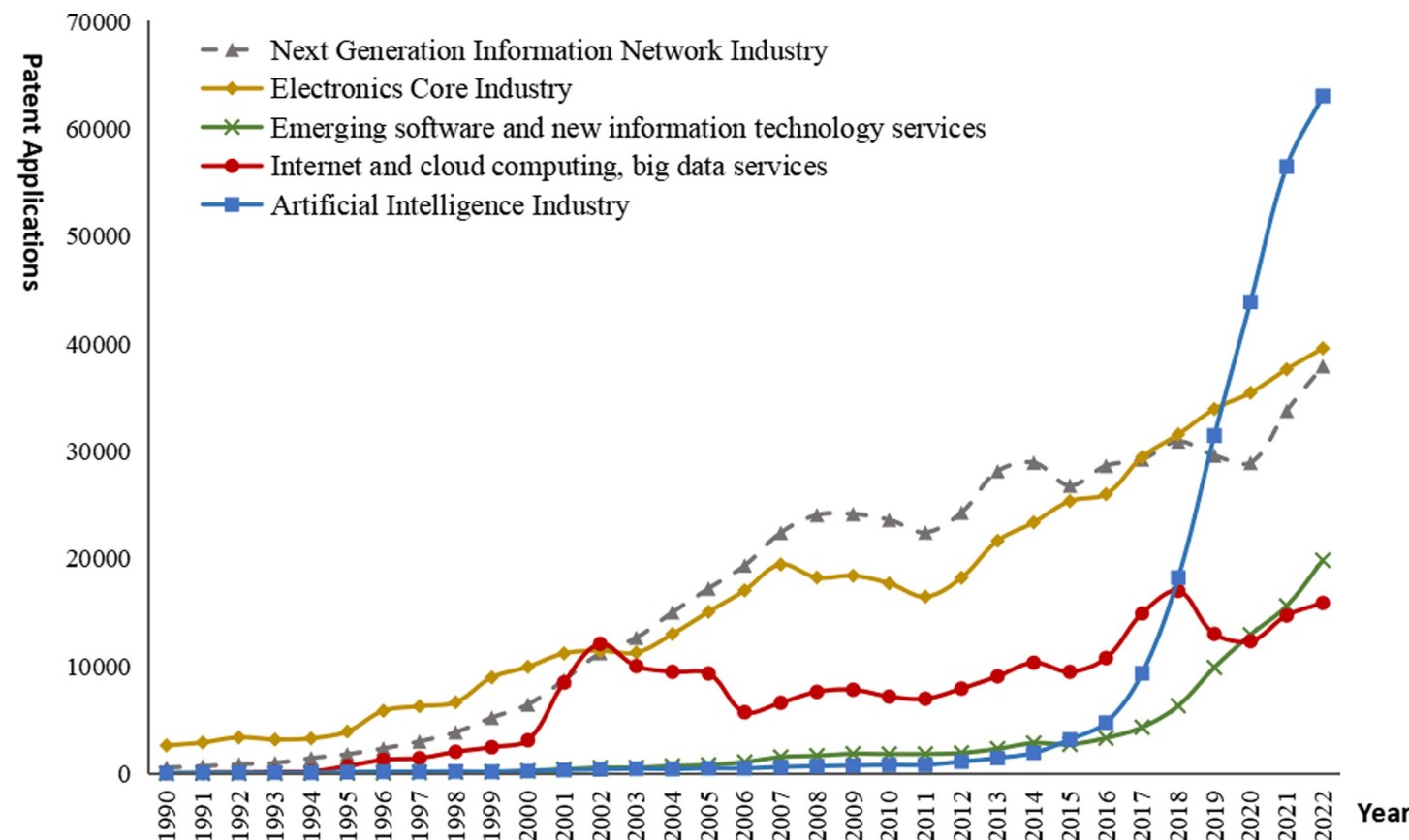

**Fig 5. The trend of patent applications of new generation emerging technology industry.**

patent applications of the Internet, cloud computing and big data services industries were flat, and related technologies were in their infancy. After 2002, the bursting of the Internet bubble hindered the development of technology to some extent. Since 2006, with the continuous deepening and improvement of Internet, cloud computing and big data technologies, patent applications have shown a steady upward trend, while the growth rate has slowed down after 2018. (5) Prior to 2006, there were only a small number of patent applications in the AI industry with a steady trend, indicating that the relevant technology was still in its early stages of development. From 2007 to 2014, the number of patent applications gradually increased, but the overall number was still small, indicating that the innovation activity of related technologies in this period was still relatively low. After 2015, as countries successively promoted the research and development of artificial intelligence technology, artificial intelligence technology made rapid breakthroughs, technology application was constantly deepened and improved, and related patent applications showed rapid and explosive growth.

## Identification of key core technology

Since the number of patents in the industry is extremely large, most of the patents have very little value. In order to reduce the calculation amount, we need to conduct preliminary screening of patents first, and exclude patents that do not meet the conditions of key core patents. According to the basic characteristics of key core technologies, the key core patents of strategic emerging industries, which make outstanding contributions to technological innovation in this field, have important influence on other patents or technological fields, and have great economic significance, and often show high citation frequencies [28]. Therefore, this paper selects the patents with the top 1000 cited frequencies and all the characteristics indexes are not 0 as the basic object of core technology identification.

According to the identification process of industrial key core technologies, the key core patents of the new generation of information technology industry segments obtained, see Table 4, the clustering results of key core technology fields are shown in Figs 6–10.

The results shown in Figs 6–10 show that:

1. The key core technologies of the next generation information network industry are wireless communication technology, communication equipment and information transmission technology. The wireless communication technology includes the H04Q-007, H04W-004, H04W-088 technical fields; communication equipment involves H04M-001, H04M-011, G06F-015, H04M-003 technical fields; information transmission technology involve H04L-012, H04L-029, H04B-007 technical fields.

2. The key core technologies of the electronic core industry are semiconductor device manufacturing methods or equipment, photoelectric technology, chemical gas processing technology. The semiconductor device manufacturing methods or equipment include H01L-021, H01L-029, H01L-027, H01L-031, H01L-023 technical fields; photoelectric technology includes G02F-001, G03F-007, G03F-009, G03B-027 technical fields; chemical gas processing technology mainly refers to C23C-016 technical field.

3. The key core technologies of emerging software and new information technology services are general data processing technology, special data processing technology, information security technology and information transmission technology. general data processing technology refers to G06F-017, G06F-015; special data processing technology involves G06Q-030 and G06Q-010; information security technology includes G06Q-020, G06F-021, H04L-009; information transmission technology refers to H04L-029, H04L-012.

**Table 4. Key core patents of new generation information technology industry.**

| Industry | Rank | Patent number | Score | Country | Rank | Patent number | Score | country |
|---|---|---|---|---|---|---|---|---|
| Next generation information network industry | 1 | WO2010036980 | 0.8260 | US | 11 | WO2008020899 | 0.2318 | US |
| | 2 | WO9934297 | 0.4973 | US | 12 | US2007047485 | 0.2283 | US |
| | 3 | WO200059196 | 0.4873 | US | 13 | WO200152567 | 0.2166 | DE |
| | 4 | WO9923844 | 0.4443 | US | 14 | WO9732439 | 0.2164 | FI |
| | 5 | WO200223362 | 0.3104 | US | 15 | WO2003058483 | 0.2126 | US |
| | 6 | WO9963709 | 0.2844 | US | 16 | WO9723108 | 0.1893 | US |
| | 7 | WO9848528 | 0.2570 | JP | 17 | WO9529536 | 0.1884 | GB |
| | 8 | WO200257865 | 0.2541 | US | 18 | WO9859447 | 0.1877 | US |
| | 9 | WO200267449 | 0.2474 | US | …… | …… | …… | |
| | 10 | WO2011085283 | 0.2362 | US | 200 | EP544449 | 0.0908 | US |
| Electronic core industry | 1 | WO2004114380 | 0.7099 | JP | 11 | EP1400859 | 0.2417 | US |
| | 2 | WO2006086423 | 0.4058 | US | 12 | WO9836888 | 0.2411 | US |
| | 3 | WO2012015550 | 0.4003 | US | 13 | EP1420299 | 0.2298 | NL |
| | 4 | WO9805078 | 0.3815 | JP | 14 | EP1486828 | 0.2194 | NL |
| | 5 | WO9852216 | 0.3293 | US | 15 | EP850920 | 0.2180 | CA |
| | 6 | WO2003104921 | 0.3249 | US | 16 | WO200161743 | 0.2126 | US |
| | 7 | EP1420298 | 0.2924 | NL | 17 | WO200215277 | 0.1986 | US |
| | 8 | WO9321748 | 0.2682 | US | 18 | WO2007146780 | 0.1776 | US |
| | 9 | WO2008143635 | 0.2577 | US | …… | …… | …… | |
| | 10 | EP1199750 | 0.2424 | US | 199 | EP1391785 | 0.0636 | NL |
| Emerging software and new information technology services | 1 | WO9745796 | 0.6719 | US | 11 | WO2009002094 | 0.2604 | US |
| | 2 | WO200143026 | 0.6543 | US | 12 | WO200052552 | 0.2592 | US |
| | 3 | WO9726061 | 0.5636 | US | 13 | WO200267175 | 0.2586 | US |
| | 4 | WO200065506 | 0.5608 | US | 14 | EP1136961 | 0.2455 | ES |
| | 5 | WO9742763 | 0.5259 | US | 15 | WO2005107410 | 0.2413 | US |
| | 6 | WO2004046875 | 0.4040 | US | 16 | WO200072124 | 0.2371 | AU |
| | 7 | WO2009111664 | 0.3178 | US | 17 | WO9924928 | 0.2215 | US |
| | 8 | WO2006108026 | 0.3135 | US | 18 | WO200188673 | 0.2161 | US |
| | 9 | JP2004357272 | 0.2720 | US | …… | …… | …… | |
| | 10 | WO200267449 | 0.2667 | US | 200 | US2012304244 | 0.0847 | US |
| Internet and cloud computing, big data services | 1 | WO200143026 | 0.6125 | US | 11 | WO200190843 | 0.2675 | US |
| | 2 | WO9727546 | 0.4834 | US | 12 | WO200031933 | 0.2635 | US |
| | 3 | WO9854581 | 0.4493 | US | 13 | WO2006073891 | 0.2537 | US |
| | 4 | WO200167303 | 0.4353 | US | 14 | EP2477421 | 0.2526 | US |
| | 5 | WO200072124 | 0.3675 | AU | 15 | WO200008909 | 0.2466 | US |
| | 6 | US2013003577 | 0.3480 | US | 16 | WO200167674 | 0.2228 | US |
| | 7 | WO200116900 | 0.3273 | US | 17 | WO2003019905 | 0.2227 | US |
| | 8 | WO200155894 | 0.3010 | US | 18 | US2005132348 | 0.2224 | US |
| | 9 | WO2006102183 | 0.2875 | US | …… | …… | …… | |
| | 10 | WO2019028269 | 0.2692 | US | 200 | WO200056033 | 0.1031 | US |

(*Continued*)

**Table 4.** (Continued）

| Industry | Rank | Patent number | Score | Country | Rank | Patent number | Score | country |
|---|---|---|---|---|---|---|---|---|
| **Artificial intelligence industry** | 1 | WO2003075129 | 0.5024 | US | 11 | WO2009059199 | 0.3369 | US |
| | 2 | WO2007103834 | 0.4196 | US | 12 | US2013080177 | 0.3346 | US |
| | 3 | WO9612256 | 0.3907 | US | 13 | US2013188040 | 0.3220 | US |
| | 4 | WO2007024573 | 0.3857 | US | 14 | WO2006081505 | 0.3162 | US |
| | 5 | EP1175060 | 0.3821 | US | 15 | US2019114511 | 0.3125 | US |
| | 6 | WO200213095 | 0.3794 | US | 16 | US2016203002 | 0.3081 | US |
| | 7 | WO2008094791 | 0.3782 | US | 17 | WO2006096208 | 0.3020 | US |
| | 8 | WO2016130719 | 0.3558 | US | 18 | WO2017128890 | 0.3019 | CN |
| | 9 | WO2004049242 | 0.3511 | US | …… | …… | …… | |
| | 10 | WO9909887 | 0.3471 | US | 200 | WO9516252 | 0.1407 | US |

4. The key core technologies of internet and cloud computing and big data services are data processing technology, data transmission technology, and digital information transmission technology. Data processing technology includes G06F-017, G06F-015, G06F-007, G06Q-030 technical fields; data transmission technology mainly refers to G06F-013, G06F-003, G06F-012 technical fields; digital information transmission technology includes H04L-012, H04M-003.

5. The key core technologies of artificial intelligence industry include intelligent control technology, identification technology, autonomous driving technology and intelligent medical technology. intelligent control technology includes G06F-003, G06F-017, G06F-015, G06F-009 technical fields; identification technology includes G06K-009, G10L-015, G06T-007,

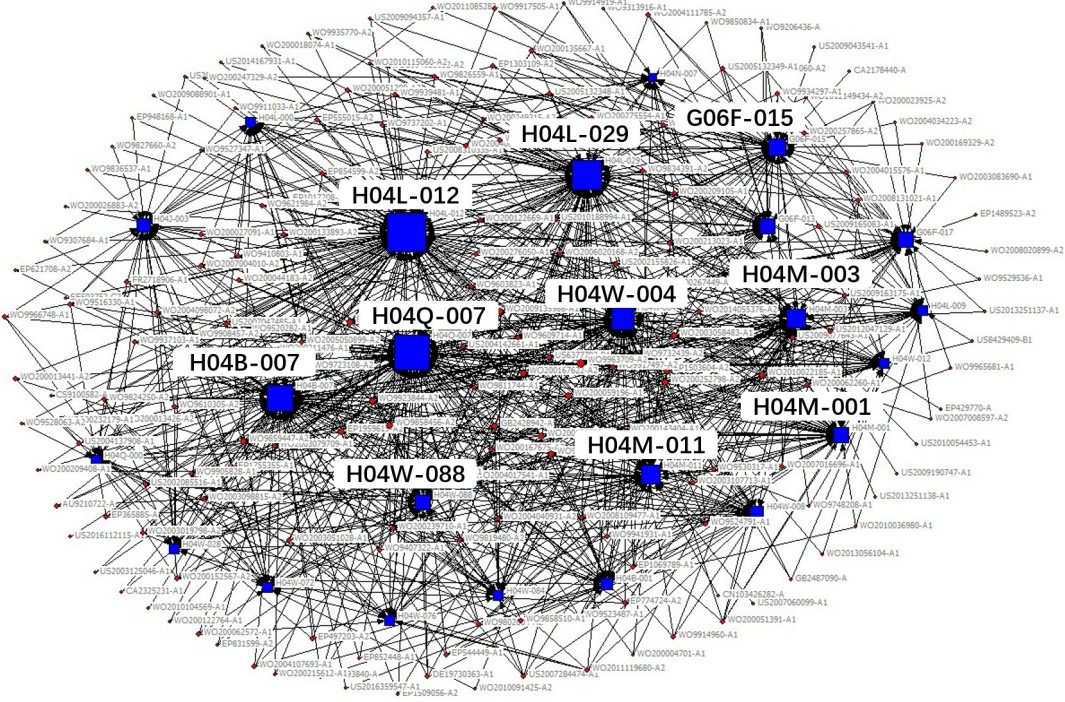

**Fig 6. Technical fields cluster chart of next generation information network industry.**

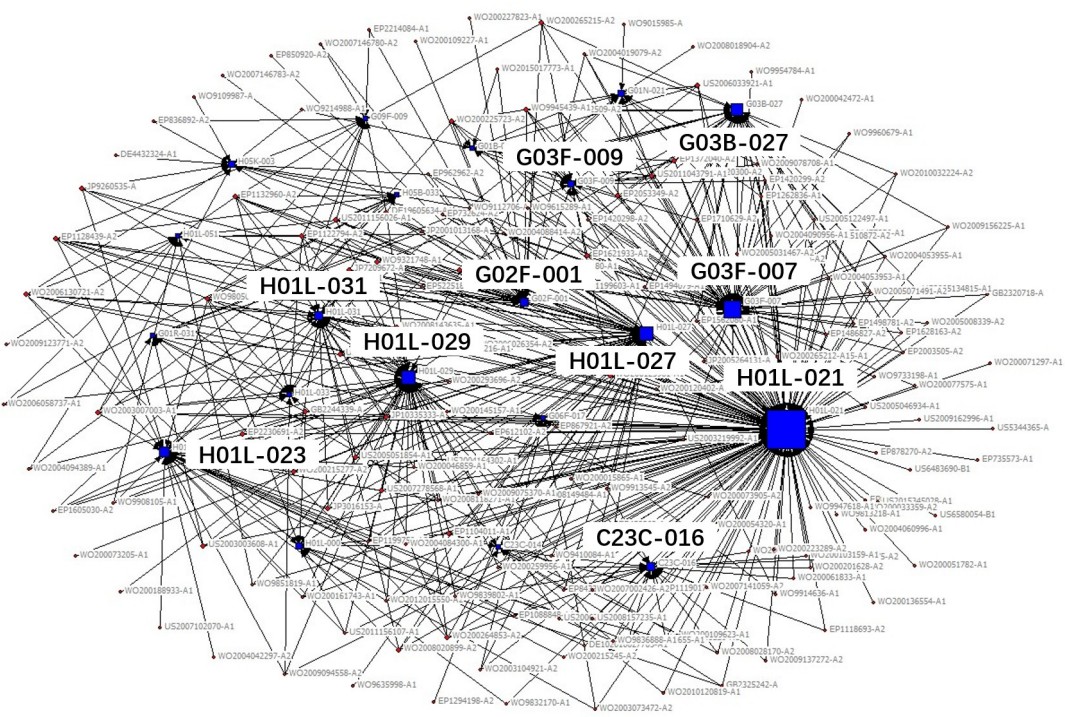

**Fig 7. Technical fields cluster chart of electronic core industry.**

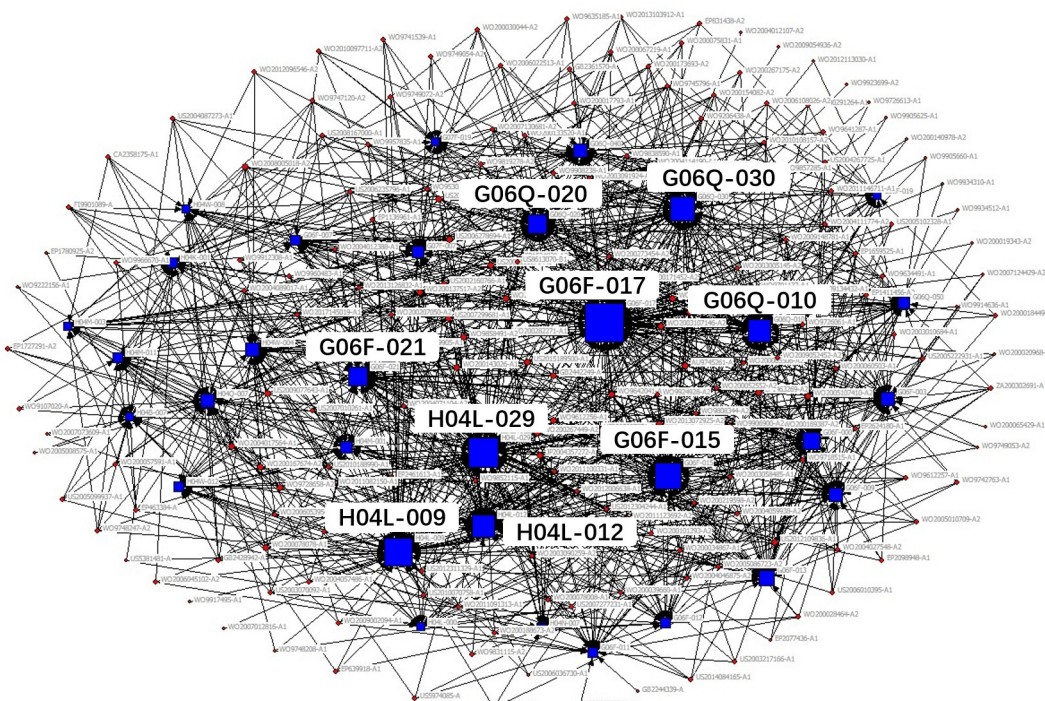

**Fig 8. Technical fields cluster chart of emerging software and new information technology services.**

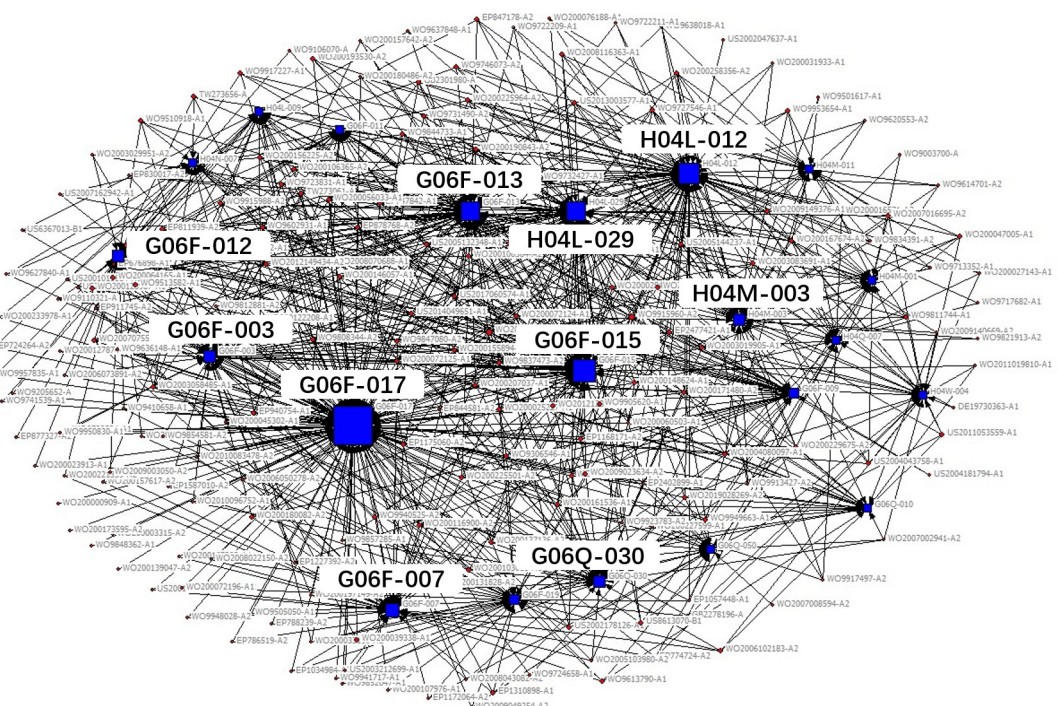

**Fig 9. Technical fields cluster chart of internet and cloud computing, big data services.**

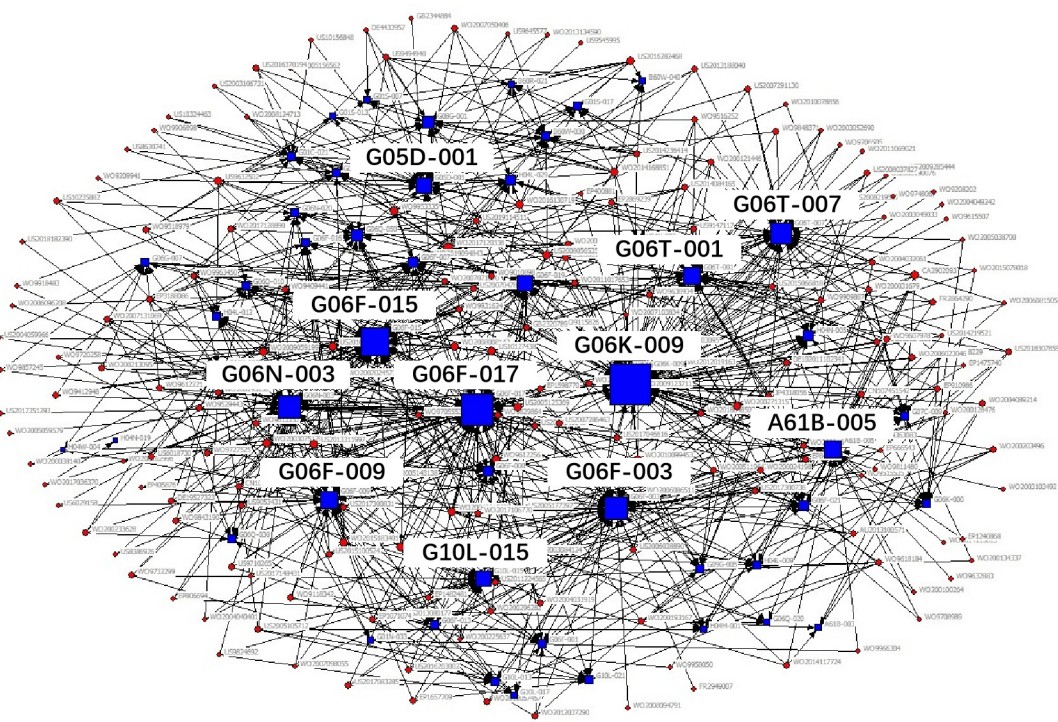

**Fig 10. Technical fields cluster chart of artificial intelligence industry.**

G06T-001, G06N-003 technical fields; autonomous driving technology mainly refers to G05D-001, and intelligent medical technology mainly refers to A61B-005 technical field.

Descriptions for specific IPC main group see S1 Appendix.

## Analysis of international competition situation of key core technologies

After sorting out the patent data of key core technology fields of each industry in the new generation of information technology industry, we obtained the index value of the international competition situation analysis of technology and draw the bubble charts, and the results are shown in Figs 11–15.

From the results displayed in the bubble charts, we found that the United States basically controls and monopolizes key core technologies in many fields of the new generation of information technology industry. Japan has strong competitiveness in many fields of new generation information networks and electronic core industries. China has been quite active in the technological activities across various fields, and has also mastered some of the key core technologies in areas such as internet and cloud computing, big data services, and AI, etc. However, there is still a certain gap between China and developed countries in other fields.

From the perspective of subdivided industries, in addition to the technology leader America, countries with strong competitiveness in key core technologies of the next generation information network industry include the developed European countries Finland, Sweden, Germany, France and the developed East Asian countries Japan and South Korea. Among them, Finland has strong competitiveness in all technical fields, and Sweden has more significant technical potential in information transmission technology, while Germany's technological expertise is primarily focused on information transmission technology, Britain has accrued certain technological proficiency in communication equipment technology, South Korea's technological competitiveness lies in the domains of information transmission and communication equipment technology, and China is moving from technology active to technology leader, with companies such as Huawei, ZTE, and China Mobile Communications possessing strong competitive advantages in information transmission and wireless communication technology.

In the electronic core industry, the United States and Japan are absolute technology leaders in various fields, and Japanese companies occupy at least half of the top ten patent patentees in each field. The Netherlands controls the key core technologies related to lithography process such as light source control, lithography process and materials, alignment or positioning, and ASML controls the world's top lithography process; South Korea is active in the field of semiconductor manufacturing methods or equipment technology activities, and Samsung has a strong competitiveness in all fields. In China, except for TSMC in Taiwan, which has excellent competitiveness in the manufacture of semiconductors and optoelectronics, mainland China lags behind other nations significantly in all areas.

In the emerging software and new information technology service industries, the United States basically controls and monopolizes the design and production of core software. IBM and Microsoft have strong technical strength. Japan and South Korea have strong competitiveness in general data processing technology and special data processing technology, and Britain, France and Germany have greater technical development potential in various fields. Australia has accumulated certain technical advantages in general data processing technology, special data processing technology and information security technology. China has a high level of R&D activity in various fields other than general data processing technology, and is gradually becoming the technology leaders, especially in the field of information transmission

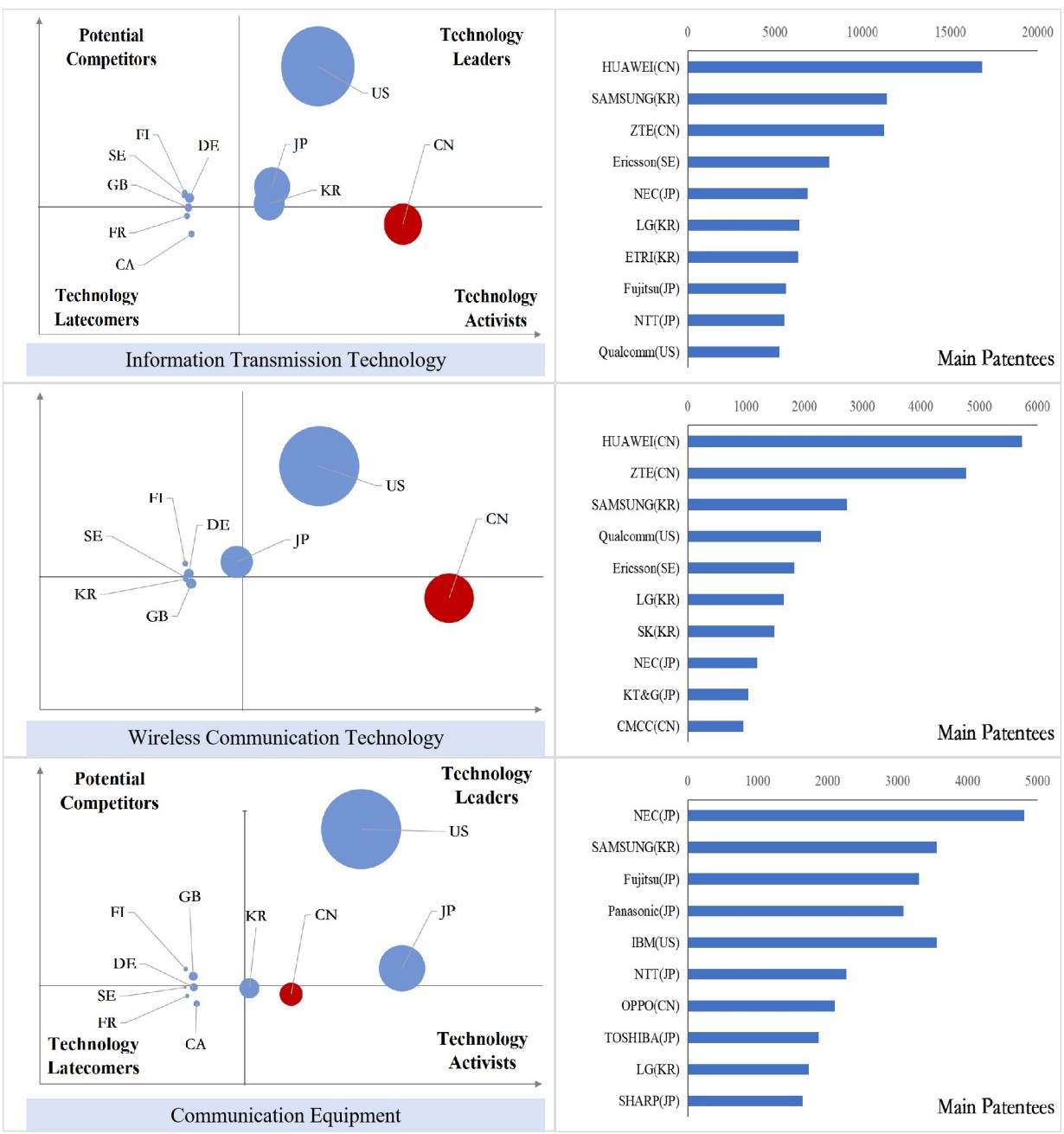

**Fig 11. Bubble chart of next generation information network industry.**

technology, Huawei, Alibaba, ZTE and other Chinese companies' technical strength is continuously emerging.

In the internet, cloud computing and big data services, except for the United States, which is the absolute technology leader, Japan has accumulated certain technological advantages in all fields; Britain and Finland have development potential in data transmission and digital information transmission; Australia's competitive advantages are reflected in data processing and data transmission; China is active in innovation activities in all fields, especially in the

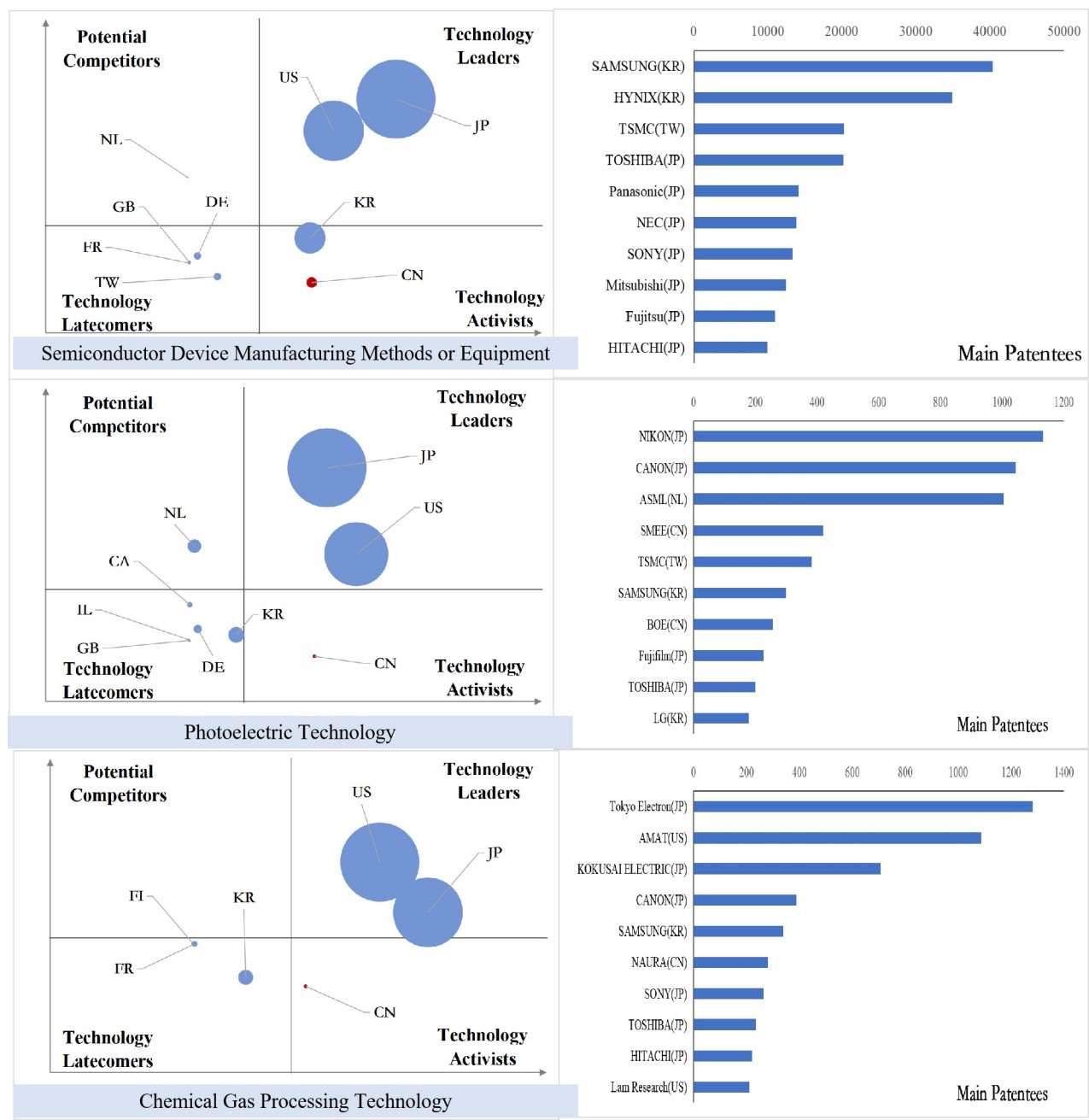

**Fig 12. Bubble chart of electronic core industry.**

field of digital information transmission technology, the global Chinese companies occupy six seats among the top ten patentees.

In the artificial intelligence industry, the United States is leaping from potential competitors to technology leaders in various fields. China is active in technological innovation activities, and has the most significant technological lead in the field of intelligent control technology, while the patent quality in other fields needs to be further improved, and patent applications are mainly from universities, indicating that the application potential of China's artificial intelligence technology needs to be further released. Japan's technology advantage is concentrated

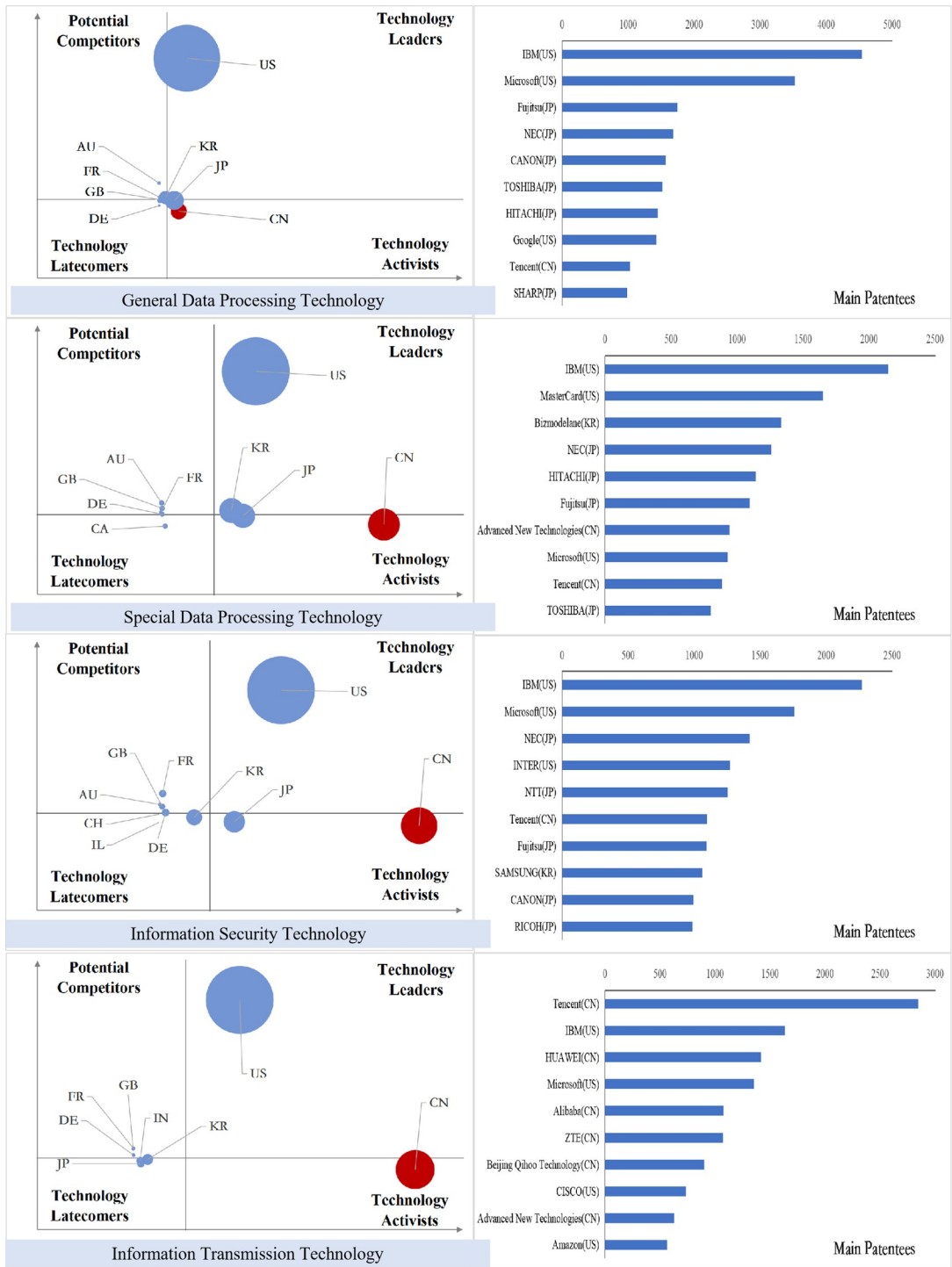

**Fig 13. Bubble chart of emerging software and new information technology services.**

in the fields of intelligent control technology and autonomous driving technology, and Germany has strong technology competitiveness in other technology fields except intelligent control technology.

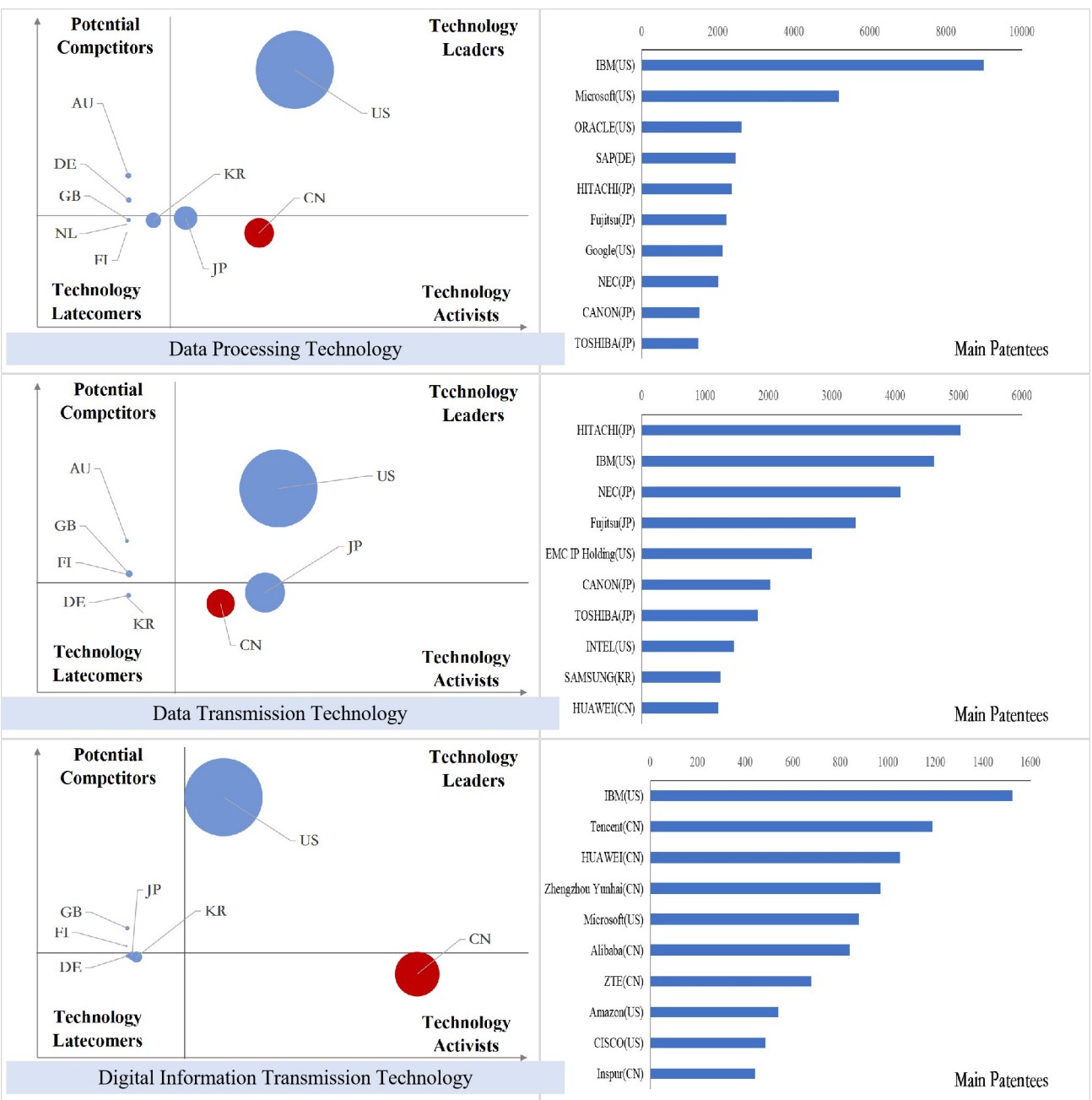

**Fig 14. Bubble chart of internet and cloud computing, big data services.**

## Conclusion

This paper constructs a framework for analyzing the international competitive situation of key core technologies in strategic emerging industries, and takes the new generation information technology industry as an example, and analyzes the key core development situation in each field based on the identification of key core patents. However, from the overall distribution of key core technologies, China's mastery of key core technologies in various fields of new-generation information technology is still low, indicating that China's underlying technologies and

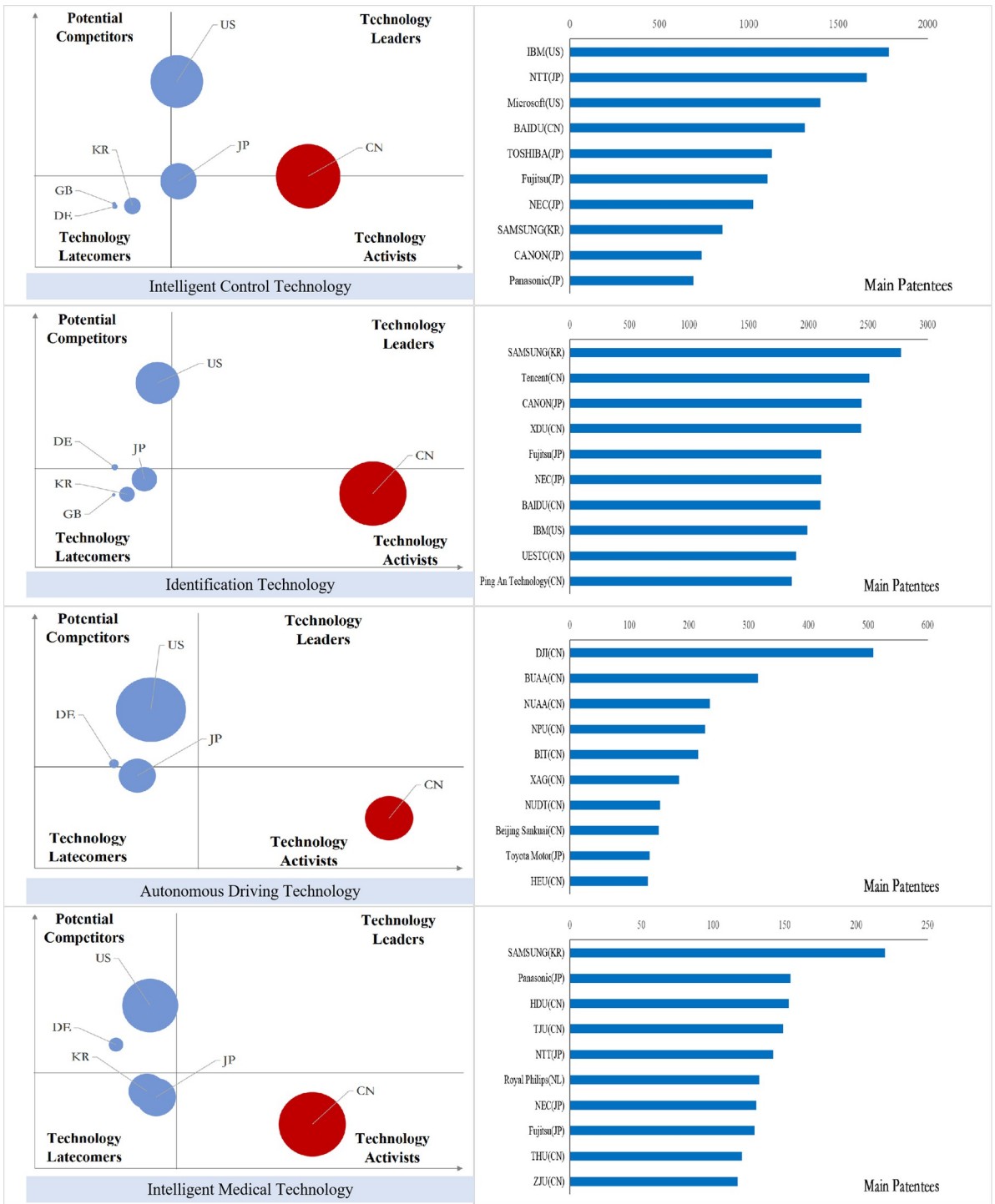

**Fig 15. Bubble chart of artificial intelligence industry.**

basic theories are still weak, and the 35 "neck" technologies of China listed by the Science and Technology Daily, including 13 new generation information technology (lithography, high-end logic chip, operating system, vacuum evaporation machine, cell phone RF devices, high-end capacitance and resistance, core industrial software, ITO targets, microspheres, database

management system, haptic sensors, LIDAR, core algorithms). Thus, the passive situation in which China's key and core technologies are dependent on and subject to other countries has not fundamentally changed. And China still has a large gap at the advanced international level, and technical bottlenecks need to be further addressed.

Under the wave of the new round of technological revolution, innovation and breakthrough of key core technologies in strategic emerging industries is a hot research issue to promote a country's high-quality development, and an important strategic issue to modernize production technology and crack the risk and hidden danger of card neck. However, the worsening situation of anti-globalization has further inhibited innovation in key core technologies in China [65], and the lack of innovation capability of key core technologies will further expand the hidden patent risks and technical difficulties faced by China in realizing industrial transformation and upgrading driven by innovation. Based on this, the following suggestions are put forward:

1. Making strategic planning at the national level. We should make overall planning and top-level design for key core technologies a priority, formulate a corresponding technology roadmap and development plan based on the international competition situation of key core technologies in various fields, and choose appropriate innovation development paths. At the same time, we should improve the accuracy of innovation policies for the research and development of key core technologies, and promote overall research and development of relevant policies and supporting measures.

2. Strengthen the synergy between industrial chains and innovation chains. We should coordinate multiple innovation subjects, improve the construction of industrial innovation systems, promote the integration of scientific research subjects with industrial development subjects, and promote the synergistic effect of scientific research and industrial development, so as to ensure the efficient allocation of innovation resources. Besides, we should try to improve the mechanism for transforming innovation results, and create a complete innovation chain of basic research—applied research—industrialization. In particular, we should vigorously consolidate the front-end basic research of key core technologies, increase the intensity of funding for basic R&D projects, promote the construction of relevant basic disciplines, and promote the integration of science and education, industry and education, so as to strengthen the supply of basic research for breakthroughs in key core technologies. Furthermore, we should make advance layout for application development fields based on the needs of industrial development, strengthen the construction of multi-disciplinary and multi-disciplinary science and technology integration, and carry out diversified cooperation with other disciplines and fields.

3. Strengthen the cultivation of short-in-supply profession talents for key core technologies breakthroughs in strategic emerging industries. Firstly, we can strengthen the construction of key talent team systems in key technical fields, and recruit leading scientific and technological talents in relevant fields globally by optimizing talent introduction policies and setting up special talent zones. Secondly, guided by the knowledge and ability needed to tackle key core technologies, we should innovate the training mechanism for high-end short-in-supply talents in majors such as integrated circuits, big data, cloud computing, Internet, blockchain and artificial intelligence. In addition, we should improve the flow mechanism between strategic emerging enterprises, universities and research institutes, and promote the orderly flow of short-in-supply profession talents in strategic emerging industries fields. Moreover, we should strengthen the awareness of openness and global vision in talents cultivation, and strengthen international exchanges and cooperation through joint training,

encouraging visiting students and studying abroad, so as to improve the ability and quality of short-in-supply talents in strategic emerging industries fields.

## Supporting information

**S1 Appendix. Descriptions for specific IPC main group.**
(PDF)

## Author Contributions

**Conceptualization:** Fengyang Wang, Zongyuan Huang.

**Data curation:** Fengyang Wang.

**Formal analysis:** Fengyang Wang, Zongyuan Huang.

**Investigation:** Fengyang Wang.

**Methodology:** Fengyang Wang.

**Supervision:** Zongyuan Huang.

**Visualization:** Fengyang Wang.

**Writing – original draft:** Fengyang Wang.

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
