## [Decision Letter · Decision Letter 0]

9 May 2023

PONE-D-23-09303Analysis of International Competitive Situation of Key Core Technology in Strategic Emerging Industries: New Generation of Information Technology Industry as an ExamplePLOS ONE

Dear Dr. Wang,

Thank you for submitting your manuscript to PLOS ONE. After careful consideration, we feel that it has merit but does not fully meet PLOS ONE’s publication criteria as it currently stands. Therefore, we invite you to submit a revised version of the manuscript that addresses the points raised during the review process.

We look forward to receiving your revised manuscript.

Kind regards,

Han Lin

Academic Editor

PLOS ONE

Journal Requirements:

2**.** PLOS requires an ORCID iD for the corresponding author in Editorial Manager on papers submitted after December 6th, 2016. Please ensure that you have an ORCID iD and that it is validated in Editorial Manager. To do this, go to ‘Update my Information’ (in the upper left-hand corner of the main menu), and click on the Fetch/Validate link next to the ORCID field. This will take you to the ORCID site and allow you to create a new iD or authenticate a pre-existing iD in Editorial Manager. Please see the following video for instructions on linking an ORCID iD to your Editorial Manager account:

Additional Editor Comments (if provided):

It might be helpful with new citations of up-to-date and high-quality papers on the subjects covered in manuscript.

Also, the author(s) should carefully format the manuscript following the guidance of the journal, such as the headers, references, and so on. There are mistakes in the citation (reference), such as missing volume, issue and page.

The authors are recommended to examine and proof the language. In addition, there are some typos and grammatical errors that need further attention.

Reviewers' comments:

Reviewer's Responses to Questions

**Comments to the Author**

1. Is the manuscript technically sound, and do the data support the conclusions?

Reviewer #1: Yes

Reviewer #2: Yes

2. Has the statistical analysis been performed appropriately and rigorously? 

Reviewer #1: Yes

Reviewer #2: Yes

3. Have the authors made all data underlying the findings in their manuscript fully available?

Reviewer #1: Yes

Reviewer #2: Yes

4. Is the manuscript presented in an intelligible fashion and written in standard English?

Reviewer #1: Yes

Reviewer #2: Yes

5. Review Comments to the Author

Reviewer #1: This paper is is very interesting. Strategic emerging industries, especially advanced information technology, are an important field of international competition. It is necessary to analyze the international competition situation of strategic emerging industries in this paper, which is of great guiding significance to the development of the new generation of information technology industries.

I think there is no problem with the paper framework, theory, method, data and conclusions. If the author can referece more papers from the international SCI or SSCI journalsa, I think it would be better.

I agree without any hesitation to accept this paper for publication.

Reviewer #2: This paper draws the conclusion that there is a gap between China and other major countries in the world, and does not put forward reasonable suggestions for China 's industrial upgrading. It is suggested to increase the relevant content of China 's industrial upgrading promotion strategy.

6. PLOS authors have the option to publish the peer review history of their article (what does this mean?). If published, this will include your full peer review and any attached files.

Reviewer #1: No

Reviewer #2: No

---

## [Author Response · Author response to Decision Letter 0]

24 May 2023

1.Respond to Editor Comments: We sincerely appreciate your careful checks. As suggested, we add more new citations of up-to-date and high-quality papers on the subjects covered in manuscript, and checked all citations. And we carefully format the manuscript following the guidance of the journal. In our resubmitted manuscript, we carefully examined and proofread the language, and typos and grammatical errors are revised. Thanks for your correction.

2.Respond to Reviewer #1: We sincerely appreciate the valuable comments. As suggested by the reviewer, we have added 29 more references from the international SCI and SSCI journals (References 1, 2, 4, 8, 10, 11, 20, 21, 22, 23, 24, 25, 26, 27, 46, 48, 49, 50, 51, 52, 54, 58, 59, 60, 61, 62, 63, 64, 65.).

3.Respond to Reviewer #2:We sincerely appreciate the valuable comments. We think this is an excellent suggestion, and we have added the relevant content at the end of the Conclusion part.

---

## [Editor Report · Decision Letter 1]

30 May 2023

Analysis of international competitive situation of key core technology in strategic emerging industries: New generation of information technology industry as an example

PONE-D-23-09303R1

Dear Dr. Huang,

We’re pleased to inform you that your manuscript has been judged scientifically suitable for publication and will be formally accepted for publication once it meets all outstanding technical requirements.

Kind regards,

Han Lin

Academic Editor

PLOS ONE
---

## [Editor Report · Acceptance letter]

5 Jun 2023

PONE-D-23-09303R1 

Analysis of international competitive situation of key core technology in strategic emerging industries: New generation of information technology industry as an example 

Dear Dr. Huang:

I'm pleased to inform you that your manuscript has been deemed suitable for publication in PLOS ONE. Congratulations! Your manuscript is now with our production department. 

Kind regards, 

on behalf of

Dr. Han Lin 

Academic Editor

PLOS ONE